# ON THE PARETO EFFICIENCY OF QUANTIZED CNNS

## ABSTRACT

Weight quantization for deep convolutional neural networks (CNNs) has shown promising results in compressing and accelerating CNN-powered applications such as semantic segmentation, gesture recognition, and scene understanding. Prior art has shown that different datasets, tasks, and network architectures admit different iso-accurate precision values, which increase the complexity of efficient quantized neural network implementations from both hardware and software perspectives. In this work, we show that when the number of channels is allowed to vary such that networks of different precision values have the same model size, lower precision values outperform higher precision ones in a Pareto sense (accuracy *vs.* model size) for networks with standard convolutions. Relying on comprehensive empirical analyses, we find that the optimal precision value of a convolution layer depends on the number of input channels per output filters and provide theoretical insights for it. To this end, we develop a simple algorithm to select the precision values for CNNs that outperforms corresponding 8-bit quantized networks by 0.9% and 2.2% in top-1 accuracy on ImageNet for ResNet50 and MobileNetV2, respectively.

## 1 INTRODUCTION

Recent success of convolutional neural networks (CNNs) in computer vision applications such as image classification and semantic segmentation, have fueled many important applications in energy-constrained devices, *e.g.*, virtual reality headsets, drones, and robots. As a result, improving the energy-efficiency of CNNs while maintaining their attractive features (*e.g.*, accuracy for a task) has gained tremendous research momentum in recent years.

Among the efforts of improving CNNs' efficiency, weight quantization was shown to be an effective technique (Zhou et al. (2016; 2017); Hou & Kwok (2018); Ding et al. (2019)). The majority of research efforts in quantization have been devoted to develop better quantization algorithms such that an iso-figure-of-merit (*i.e.*, accuracy) is achieved with lowest possible weight precision value. Nevertheless, the iso-accurate precision value depends on the dataset, task, and network architecture of interest, which greatly increases the neural network implementation complexity from both hardware and software perspectives. For example, hardware and software implementations optimized for executing an 8 bit network are sub-optimal for executing a 4 bit network, and vice versa. The design optimization complexity further increases for recently proposed mixed-precision networks (Wang et al. (2019); Wu et al. (2018b); Dong et al. (2019)).

The key observation we have is that most prior literature in this space studies quantization for fixed network architectures, which is reasonable for evaluating the effectiveness of quantization algorithms, but unnecessary when considering the Pareto efficiency (accuracy *vs.* model size) of neural networks. In this work, we relax the restriction of fixing the network architecture and allow the number of channels of the CNN under consideration to vary. More concretely, we use the width-multiplier[1] (Howard et al. (2017)) as a tool to compare the performance of different weight precision values under *the same model size*.

Overall, we systematically analyze the model size and accuracy trade-offs considering both weight precision values and the number of channels for various modern networks architectures (variants of

---

[1]Width-multiplier grows or shrinks the number of channels across the layers with identical proportion for a certain network, *e.g.*, grow the number of channels for all the layers by $2\times$.

ResNet, VGG, and MobileNet) and datasets (CIFAR and ImageNet) and have the following non-trivial and novel contributions:

- We are the first to empirically show that when considering channel counts, lower precision weight values outperform higher precision weight values in a Pareto sense (accuracy vs. model size) for networks with standard convolutions. This is intriguing since it implies that scaling up (in terms of model size) along the channel count dimension is more effective for accuracy than the precision value dimension.

- We are the first to show that the fan-in channel counts per output filter for a convolution layer determine the effectiveness of accuracy improvement when the model is scaled along the weight precision dimension and provide both theoretical and empirical reasoning for this.

- We are the first to show that with a simple model scaling rule (the proposed *DualPrecision*), one can achieve a more accurate model (given the same model size) even compared to mixed-precision prior art that uses deep reinforcement learning to search for layer-wise weight precision values. Moreover, the results are validated on the large-scale dataset, *i.e.*, ImageNet. This is a manifestation of our two previous findings.

The remainder of the paper is organized as follows. Section 2 discusses related work. Section 3 discusses the methodology used to discover our findings. Section 4 shows lower precision values are preferable for networks with standard convolutions. Section 5 discusses how fan-in channel count per output filter affects precision scaling for convolution layers. Section 6 discusses *DualPrecision*, our simple yet effective model scaling rule. Section 7 concludes the paper.

## 2 RELATED WORK

Several techniques for improving the efficiency of CNNs have been recently proposed. For instances, pruning removes the redundant connections of a trained neural network (Zhuang et al. (2018); Ye et al. (2018); Theis et al. (2018); Li et al. (2017); Frankle & Carbin (2019); Chin et al. (2019); Yu et al. (2018)), neural architecture search (NAS) tunes the number of channels, size of kernels, and depth of a network (Tan et al. (2018); Stamoulis et al. (2019); Cai et al. (2018); Stamoulis et al. (2018)), and convolution operations can be made more efficient via depth-wise convolutions (Howard et al. (2017)), group convolutions (Huang et al. (2017); Zhao et al. (2019b)), and shift-based convolutions (He et al. (2019); Wu et al. (2018a)). In addition to the aforementioned techniques, network quantization introduces an opportunity for hardware-software co-design to achieve better efficiency for CNNs.

There are in general two directions for weight quantization in prior literature, post-training quantization (Nagel et al. (2019); Meller et al. (2019); Zhao et al. (2019a); Sheng et al. (2018)) and quantization-aware training (Rastegari et al. (2016); Zhu et al. (2017); Jacob et al. (2018); Jung et al. (2019); Yuan et al. (2019); Hou & Kwok (2018); Choi et al. (2018)). The former assumes training data is not available when quantization is applied. While being fast and training-data-free, its performance is worse compared to quantization-aware training. In contrast, our work falls under the category of quantization-aware training.

In quantization-aware training, (Rastegari et al. (2016)) introduces binary neural networks, which lead to significant efficiency gain by replacing multiplications with XNOR operations, but at the expense of significant accuracy degradation. Later, (Zhu et al. (2017)) propose ternary quantization and (Zhou et al. (2016); Jacob et al. (2018)) bridge the gap between floating-point networks and binarized neural networks by introducing fixed-point quantization. Building upon prior art, the vast majority of existing work focuses on reducing the accuracy degradation by improving the training strategy (Zhou et al. (2017); Yang et al. (2019); Louizos et al. (2019); Ding et al. (2019)) and better quantization schemes (Jung et al. (2019); Wang et al. (2019); Yuan et al. (2019)). However, prior art studies quantization by fixing the network architecture, which may lead to sub-optimal precision decisions in terms of Pareto efficiency (model size *vs.* accuracy).

Related to our work, Mishra et al. (2018) have also considered the impact of channel count in quantization. In contrast, our work has the following novel features. First, we find that in CNNs with standard convolutions, *lower precision values outperform higher ones in a Pareto sense*. Second, we

find that the Pareto optimal precision value depends on the number of input channels per output filter and provide theoretical insights for it. Last, we propose an algorithm to select the precision values of a given network which as a result outperforms *8 bit fixed-point* and *mixed-precision* baselines.

# 3 METHODOLOGY

We conduct all of our experiments on image classification datasets including CIFAR-100 and ImageNet. All the experiments are trained from scratch to ensure different precision values are trained equally long. While we do not start from a pre-trained model, we note that our baseline fixed-point models (*i.e.*, 4 bit for CIFAR and 8 bit for ImageNet) achieve iso-accurate results compared to their floating-point counterparts. For all the experiments on CIFAR, we run the experiments three times and report the mean and standard deviation. The training hyper-parameters are detailed in Appendix A.

## 3.1 QUANTIZATION

While our work focuses on weight quantization, we still quantize the activations since they are normally quantized for efficient deployment (Jacob et al. (2018)). For activation quantization, we use the technique proposed in prior art (Jacob et al. (2018)) and use 4 bit for CIFAR-100 and 8 bit for ImageNet experiments. We note that the precision value is chosen such that iso-accurate results can be achieved when compared to the floating-point baselines.

For weight quantization, we use a straight-through estimator (Bengio et al. (2013)) to conduct quantization-aware training. Specifically, for precision beyond 2 bit ($b > 2$), we use the following quantization function for weights during the forward pass:

$$Q(\boldsymbol{W}_{i,:}) = \lfloor \frac{clamp(\boldsymbol{W}_{i,:}, -a_i, a_i)}{s_i} \rceil \times s_i, \;\; s_i = \frac{a_i}{2^{b-1} - 1}, \tag{1}$$

where $\lfloor \cdot \rceil$ denotes the round-to-nearest-neighbor function, $\boldsymbol{W} \in \mathbb{R}^{C_{out} \times d}$, $d = C_{in} K_w K_h$ denotes the real-value weights for the $i^{\text{th}}$ output filter of a convolution layer that has $C_{in}$ channels and $K_w \times K_h$ kernel size. $\boldsymbol{a} \in \mathbb{R}^{C_{out}}$ denotes the vector of clipping factors which are selected to minimize $\|Q(\boldsymbol{W}_{i,:}) - \boldsymbol{W}_{i,:}\|_2^2$ by assuming $\boldsymbol{W}_{i,:} \sim \mathcal{N}(0, \sigma^2 \boldsymbol{I})$. More details about the determination of $a_i$ is in Appendix B.

For special cases such as 2 bit and 1 bit, we use schemes proposed in prior art. Specifically, let us first define:

$$|\bar{\boldsymbol{W}}_{i,:}| = \frac{1}{d} \sum_{j=1}^{d} |\boldsymbol{W}_{i,j}|. \tag{2}$$

For 2 bit precision, we follow trained ternary networks (Zhu et al. (2017)) and define the quantization function as follows:

$$Q(\boldsymbol{W}_{i,:}) = (sign(\boldsymbol{W}_{i,:}) \odot \boldsymbol{M}_{i,j}) \times (|\bar{\boldsymbol{W}}_{i,:}|)$$
$$\boldsymbol{M}_{i,j} = \begin{cases} 0, & \boldsymbol{W}_{i,j} < 0.7|\bar{\boldsymbol{W}}_{i,:}|. \\ 1, & otherwise. \end{cases} \tag{3}$$

For 1 bit precision, we follow DoReFaNets (Zhou et al. (2016)) and define the quantization function as follows:

$$Q(\boldsymbol{W}_{i,:}) = sign(\boldsymbol{W}_{i,:}) \times (|\bar{\boldsymbol{W}}_{i,:}|). \tag{4}$$

For the backward pass for all the precision values, we use a straight-through estimator as in prior art to make the training differentiable. That is,

$$\frac{Q(\boldsymbol{W}_{i,:})}{\partial \boldsymbol{W}_{i,:}} = \boldsymbol{I}. \tag{5}$$

In the sequel, we quantize the *first and last layers to 8 bit*. They are fixed throughout the experiments. We note that it is a common practice to leave the first and the last layer *un-quantized* (Zhou et al. (2016)), however, we find that using 8 bit can achieve iso-accurate results.

### 3.2 COST METRICS

To measure the cost of CNN models, we use the size of the model ($C_{size}$) defined as:

$$\mathcal{C}_{size} = \sum_{i=1}^{O} b(i)C_{in}(i)K_w(i)K_h(i) \tag{6}$$

where $O$ denotes the total number of filters and $b(i)$ denotes the precision for filter $i$, $C_{in}(i)$ denotes the number of channels for filter $i$, and $K_w(i)$ and $K_h(i)$ denote the kernel height and width for filter $i$. We choose model size as a metric because it is relevant to both machine learning and systems community. Specifically, model size is of interest for the machine learning community since it represents a proxy of model complexity. On the other hand, for the systems community, model size is related to latency and energy for CNNs with weight fetch dominating memory accesses (*e.g.*, the streaming inference scenario where inference is done with single data instance per batch). We note that this metric is also adopted in the MicroNet Challenge (Gale et al. (2019)) held at NeurIPS 2019.

## 4 DIFFERENT PRECISION VALUES HAVE DIFFERENT PARETO EFFICIENCY

To be precise in the following discussion, we define *Pareto domination* as follows:

**Definition 4.1 (Pareto domination)** *When comparing precision values A and B for a network family $\mathcal{F}$, we say A **Pareto dominates** B if,*

$$Acc(N(A, s)) > Acc(N(B, s)) \ \forall s,$$

*where Acc evaluates the validation accuracy of a network, $N(A, s)$ uses width-multiplier to find a network in $\mathcal{F}$ such that it has A precision value and s model size.*

We study three kinds of commonly adopted CNNs, namely, ResNets with Basic Block (He et al. (2016)), VGG (Simonyan & Zisserman (2014)), and MobileNetV2 (Sandler et al. (2018)). These networks differ in the convolution operations, connections, and filter counts. For ResNets, we explored the network from 20 layers up to 56 layers in a step of six layers. For VGG, we investigate VGG with eleven layers. Additionally, we also study MobileNetV2, which is a mobile-friendly network. We note that we modify the stride count in of the original MobileNetV2 to match the number of strides of ResNet for CIFAR. The used architectures are discussed in detail in Appendix C.

For CIFAR-100, we only study precision values below 4 bit since the latter can achieve iso-accurate results compared to its floating-point counterpart. Specifically, we consider 4 bit, 2 bit, and 1 bit precision values. To compare the Pareto efficiency of different precision values, we use the width-multiplier to align the model size among them. For example, one can make a 1-bit CNN 2× wider to align with the model size of a 4-bit CNN [2]. For each network, we sweep the width-multiplier to consider points at multiple model sizes. As it can be observed from Figure 1, across the three types of networks we study, there exists some precision value that Pareto dominates others. For ResNets and VGG, it is 1 bit. In contrast, for MobileNetV2, it is 4 bit. The results for ResNets and VGG are particularly interesting, since we observe that the lower precision value Pareto dominates the higher precision ones. This implies that for networks such as ResNets and VGG, scaling the model along the channel dimension is always more preferable in accuracy-vs-size trade-off compared to scaling the model along the weight precision value dimension.

## 5 THE OPTIMAL PRECISION VALUE DEPENDS ON THE NUMBER OF FAN-IN CHANNELS

With the empirical results from Section 4, we have learned that lower precision values are better for two of the networks we study but not for MobileNetV2, which has a reversed behavior. In this section, we are interested in identifying the underlying cause for this different trend. Through a

---

[2]Increase the width of a layer increases the number of output filters for that layer as well as the number of channels for the subsequent layer. Thus, number of parameters and number of operations grow approximately quadratically with the width-multiplier.

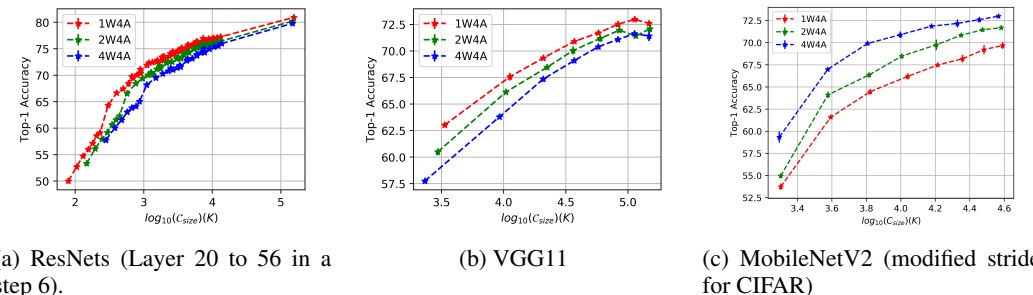

(a) ResNets (Layer 20 to 56 in a step 6).

(b) VGG11

(c) MobileNetV2 (modified stride for CIFAR)

Figure 1: Comparisons of the Pareto efficiency for different precision under three kinds of CNNs. $x$W$y$A denotes $x$-bit weight quantization and $y$-bit activation quantization. The experiments are done on the CIFAR-100 dataset. For each network, we sweep the width-multiplier to cover points at multiple model sizes.

series of controlled experiments, we empirically identify that more channels per output filter leads to lower optimal precision value. In addition, we provide theoretical insights behind this empirical result.

## 5.1 DEPTH-WISE CONVOLUTION

As it can be observed in Figure 1, MobileNetV2 is a special case where higher precision values Pareto dominate lower ones. When comparing MobileNetV2 to the other two networks, there are many differences, including how convolutions are connected, how many convolution layers are there, how many filters in each of them, and how many channels for each convolution. To narrow down which of these impacts the reversed trend, we first consider the inverted residual blocks, *i.e.*, the basic component in MobileNetV2. To do so, we replace all basic blocks (two consecutive convolutions) of ResNet26 with the inverted residual blocks. We refer to this new network as Inv-ResNet26. As

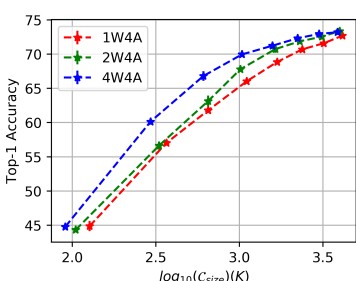

Figure 2: The trade-off curves for Inv-ResNet26.

shown in Figure 2, the Pareto efficiency trend of Inv-ResNet26 resembles the one of MobileNetV2 and recall that in case of ResNet26, lower precision values Pareto dominate higher ones. Thus, we can infer that the inverted residual block itself or its components are responsible for such a reversed trend.

Since an inverted residual block is composed of a point-wise convolution and a depth-wise separable convolution, we further consider the case of depth-wise separable convolution (DWSConv). To identify whether DWSConv can cause the trend reversion, we use VGG11 as a starting point

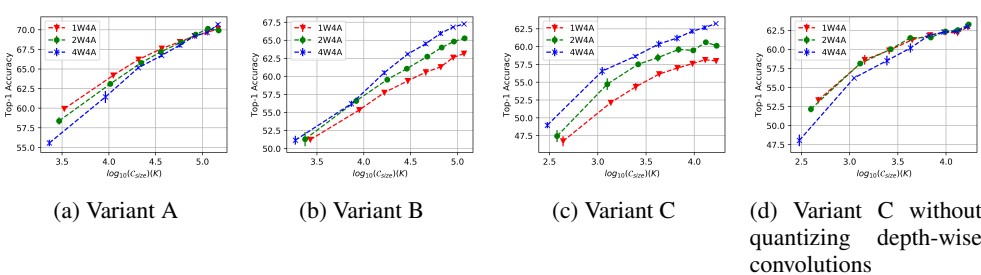

(a) Variant A

(b) Variant B

(c) Variant C

(d) Variant C without quantizing depth-wise convolutions

Figure 3: The trade-off curves for the three variants of VGG11 that we investigated.

and gradually replace each of the convolution with DWSConv. We note that replacing all convolutions with DSWConvs results in an architecture that resembles MobileNetV1 (Howard et al. (2017)). Specifically, we introduce three variants of VGG11 that have an increasing number of convolutions replaced by DWSConvs. Starting with the second layer, *variant A* has one layer replaced by DWS-Conv, *variant B* has four layers replaced by DWSConvs, and *variant C* has all of the layers except for the first layer replaced by DWSConvs (the architectures are detailed in Appendix C).

As shown in Figure 3, as the number of DWSConv increases (from variant A to variant C), the optimal precision value shifts from 1 bit to 4 bit, which implies that depth-wise separable convolutions or the layers within it are affecting the optimal precision value. To identify which of the layers of the DWSConv (*i.e.*, the depth-wise convolution or the point-wise convolution) is more important in affecting the optimal precision value, we keep the precision value of depth-wise convolutions fixed at 4 bit and quantize other layers. As shown in Figure 3d, the optimal curve shifts from 4 bit being the best back to 1 bit, with a similarly performing 2 bit. Thus, depth-wise convolutions appear to directly affect the optimal precision trends.

## 5.2 SENSITIVITY ANALYSIS

In our setup, to obtain a lower precision network that has the same model size as a higher precision network we follow two steps: (1) quantize the network weights to lower-precision values and (2) grow the network with width-multiplier to the model size of the higher-precision one. The two steps introduce accuracy differences of $\Delta Acc_Q = Acc_{low} - Acc_{high}$ and $\Delta Acc_G = Acc_{low,grown} - Acc_{low}$, respectively. Since depth-wise convolutions introduce a reverse trend in Pareto efficiency, which is the result of $\Delta Acc_Q + \Delta Acc_G$, the reason can potentially be due to them being quantization-unfriendly, growing-unfriendly, or both.

To further diagnose the reason why depth-wise convolutions have a reverse Pareto efficiency trend, we analyze the accuracy differences for networks with and without quantizing depth-wise convolutions, *i.e.*, Figure 3c and Figure 3d. Specifically, we use width-multipliers of $1\times$, $1.25\times$, $1.5\times$, $1.75\times$, and $2\times$ for the 4-bit variant C as networks of higher precision. Thus, $\Delta Acc_Q$ is evaluated against the corresponding 1-bit quantized model and $\Delta Acc_G$ is measured by comparing the 1-bit model and its $2\times$ grown counterpart. As shown in Table 1, when quantizing depth-wise convolutions, $\Delta Acc_Q$ becomes more negative such that $\Delta Acc_Q + \Delta Acc_G < 0$. This implies that the main reason for the optimal precision value change is that depth-wise convolutions are quantization-unfriendly when going below 4 bit. We note that we expected that quantizing the depth-wise convolutions would incur smaller $\Delta Acc_Q$ compared to their no-quantization baseline because we essentially quantized more layers. However, depth-wise convolutions only account for 2% of the model size but incur on average near $4\times$ more accuracy degradation when quantized.

We note that Sheng et al. (2018) also find that depth-wise separable convolutions are quantization-unfriendly. However, their results are based on post-training layer-wise quantization. As mentioned in their work (Sheng et al. (2018)), the quantization challenges in their setting could be resolved by quantization-aware training, which is the scheme considered in this work. As a result, our finding is different and novel.

## 5.3 QUANTIZATION AND DEPTH-WISE CONVOLUTIONS

Having uncovered that depth-wise convolutions introduce large accuracy degradation when weights are quantized below 4 bit, in this section, we investigate depth-wise convolutions from a quantization perspective. When comparing depth-wise convolutions and standard convolutions in the context of

Table 1: Analysis of the impact of depth-wise convolutions with $\Delta Acc_Q = Acc_{1bit} - Acc_{4bit}$ and $\Delta Acc_G = Acc_{1bit,2\times} - Acc_{1bit}$ by comparing variant C with and without quantizing the depth-wise convolutions from 4 bit to 1 bit.

| VARIANT C | 1.00× | | 1.25× | | 1.50× | | 1.75× | | 2.00× | | AVERAGE | |
|---|---|---|---|---|---|---|---|---|---|---|---|---|
| | $\Delta Acc_Q$ | $\Delta Acc_G$ | $\Delta Acc_Q$ | $\Delta Acc_G$ | $\Delta Acc_Q$ | $\Delta Acc_G$ | $\Delta Acc_Q$ | $\Delta Acc_G$ | $\Delta Acc_Q$ | $\Delta Acc_G$ | $\Delta Acc_Q$ | $\Delta Acc_G$ |
| 4 BIT DWCONV | -1.54 | +2.61 | -2.76 | +2.80 | -1.77 | +1.74 | -1.82 | +1.64 | -1.58 | +1.55 | -1.89 | +2.07 |
| QUANT. ALL | -8.60 | +4.39 | -7.60 | +3.41 | -7.74 | +3.19 | -8.61 | +4.09 | -7.49 | +2.25 | -8.01 | +3.47 |

quantization, they differ in the number of elements to be quantized, *i.e.*, $C_{in} = 1$ for depth-wise convolutions and $C_{in} >> 1$ for standard convolutions.

Why does the number of elements matter? In quantization-aware training, one needs to estimate some statistics of the vector to be quantized (*i.e.*, $a$ in Equation 1 and $|\bar{w}|$ in Equations 3,4) based on the elements in the vector. The number of elements affect the robustness of the estimate that further decides the quantized weights. More formally, we provide the following proposition.

**Proposition 5.1** *Let $w \in \mathbb{R}^d$ be a the weight vector to be quantized where $w_i$ has distribution of $\mathcal{N}(0, \sigma^2) \; \forall \; i$ without assuming samples are drawn independently and $d = C_{in}K_wK_h$. If the average correlation of the weights is denoted by $\rho$, the variance of $|\bar{w}|$ can be written as follows:*

$$\text{Var}(|\bar{w}|) = \frac{\sigma^2}{d} + \frac{(d-1)\rho\sigma^2}{d} - \frac{2\sigma^2}{\pi}. \tag{7}$$

The proof is in Appendix D. This proposition states that, as the number of elements ($d$) increases, the variance of the estimate can be reduced due to the first term. The second term depends on the correlation between weights. Since the weights might not be independent during training, the variance is also affected by their correlations.

We empirically validate Proposition 5.1 by looking into the sample variance of $|\bar{w}|$ across the course of training[3] for different $d$ values by growing $(K_w, K_h)$ or $C_{in}$. To do so, we consider the $0.5\times$ VGG variant C by changing the number of elements of the depth-wise convolutions. Since $d = (C_{in} \times K_w \times K_h)$ for a convolution layer, we consider the original depth-wise convolution, *i.e.*, $d = 1 \times 3 \times 3$ and increasing channels with $d = 4 \times 3 \times 3$ and $d = 16 \times 3 \times 3$, and increasing kernel size with $d = 1 \times 6 \times 6$, , and $d = 1 \times 12 \times 12$. The numbers are selected such that growing the channel has the same $d$ for the corresponding higher kernel size.

In Figure 4, we analyze the layer-level sample variance by averaging it for all the filters in the same layer. First, we observe that one can reduce the variance by increasing the number of elements along both the channel and kernel size dimensions. Second, we find that increasing the number of channels is more effective than increasing the kernel size in reducing the variance, which could be due to a different correlation of the weights, *i.e.*, intra-channel weights have larger correlation than inter-channel weights.

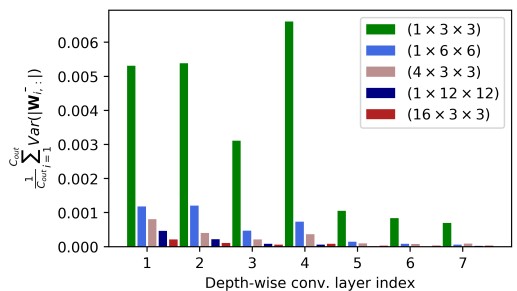

Figure 4: The average estimate $\text{Var}(|\bar{w}|)$ for each depth-wise convolution under different $d = (C_{in} \times K_w \times K_h)$ values.

However, lower variance might not necessarily imply lower quantization error for the quantized models. Thus, we conduct the $\Delta Acc$ analysis for different $d$ values. More specifically, we want to understand how $d$ affects the accuracy difference between lower precision (1 bit) and higher precision (4 bit) models ($\Delta Acc_Q$) and the accuracy difference between the lower precision (1 bit) and its grown ($2\times$) counterpart ($\Delta Acc_G$). As shown in Table 2, we empirically find that lower variance reflects larger $\Delta Acc_Q$ (less degradation). On the other hand, when comparing channel counts and kernel sizes, we observe that increasing the number of channels is more effective than increasing the kernel size in reducing accuracy degradation (larger $\Delta Acc_Q$). Moreover, we find that increasing kernel size reduces $Acc_G$ more than increasing the number of channels; this may be because a larger kernel is harder to optimize and the CIFAR dataset does not benefit from larger receptive field. Indeed, from the last row of Table 2, we can observe that increasing the kernel size reduces the accuracy for the 4 bit models.

Overall, from the Pareto efficiency perspective, we are interested in $\Delta Acc$, which determines whether the lower precision can have better accuracy when grown to the same model size as the

---

[3]We treat the calculated $|\bar{w}|$ at each training step as a sample and calculate the sample variance across training steps.

Table 2: Analysis of the impact of $d$ with $\Delta Acc_Q$ and $\Delta Acc_G$ for VGG variant C. $\Delta Acc_Q = Acc_{1bit} - Acc_{4bit}$ and $\Delta Acc_G = Acc_{1bit,2\times} - Acc_{1bit}$.

| Variant C | $(1 \times 3 \times 3)$ | $(1 \times 6 \times 6)$ | $(4 \times 3 \times 3)$ | $(1 \times 12 \times 12)$ | $(16 \times 3 \times 3)$ |
|---|---|---|---|---|---|
| $\Delta Acc_Q$ | $-12.37 \pm 1.16$ | $-9.21 \pm 0.68$ | $-6.42 \pm 0.41$ | $-7.07 \pm 0.16$ | $-4.45 \pm 0.49$ |
| $\Delta Acc_G$ | $8.72 \pm 0.59$ | $6.79 \pm 0.88$ | $7.20 \pm 0.30$ | $4.49 \pm 0.45$ | $5.40 \pm 0.66$ |
| $\Delta Acc$ | $-3.66 \pm 0.58$ | $-2.41 \pm 0.27$ | $0.79 \pm 0.13$ | $-2.58 \pm 0.33$ | $0.95 \pm 0.39$ |
| $Acc_{4bit}$ | $56.24 \pm 0.37$ | $55.63 \pm 0.47$ | $58.51 \pm 0.39$ | $53.31 \pm 0.22$ | $60.97 \pm 0.30$ |

higher precision model. In this case, we find empirically that, as the number of channels per output filter increases, $\Delta Acc$ increases. This implies that *higher fan-in channel counts per output filter can benefit more from using lower weight precision values.*

## 6 DUALPRECISION: PRECISION SELECTION FOR CNNS

From previous results, we find that the optimal precision value depends on the number of fan-in channels per output filter in a convolution layer and as the number of fan-in channels grows, the optimal precision value becomes smaller. Together with the observation that convolution layers in modern CNNs, except for depth-wise convolutions, have many channels per filter, we propose *DualPrecision*, which uses one precision value (presumably higher) for depth-wise convolutions and another precision value (presumably lower) for other convolution layers. Once the precision values are found, we use width-multipliers to grow or shrink the network to the desired model size.

With this heuristic, the search space of precision selection becomes so small that grid search is feasible, *i.e.*, $|\mathbb{B}| \times |\mathbb{B}|$ for networks with depth-wise convolutions and $|\mathbb{B}|$ otherwise. $\mathbb{B}$ denotes the set of considered precision values and is typically small, *e.g.*, $\{1, 2, 4, 8\}$. We note that the search space for mixed precision (Wang et al. (2019); Wu et al. (2018b)) is $|\mathbb{B}|^L$ with $L$ being the number of layers. In *DualPrecision*, one can explore the grid more efficiently by using heuristics that incorporate our findings. Specifically, we find that one precision value Pareto dominates others and as a result, one can compare precision values at the regime of low computational cost so as to train the network faster.

We evaluate the proposed *DualPrecision* with ResNet50 and MobileNetV2 on the ImageNet dataset. Since we keep the precision of the first and last layer quantized at 8 bit, scaling them in terms of width will grow the number of parameters much more quickly than other layers. As a result, we keep the number of channels for the first and last channel fixed for the ImageNet experiments. We first conduct grid search ($|\mathbb{B}| = \{1, 2, 4, 8\}$) for ResNet50 and MobileNetV2 by scaling them down with width-multipliers so as to make the grid search faster. Once the optimal precision is decided, we use width-multipliers to traverse the trade-off curve. Specifically, we use the model size of the $0.25\times$ 8-bit model to conduct grid search for both networks. For ResNet50, there are only four precision values to be searched while MobileNetV2 has 16 such values. The grid search results are shown in Appendix E.

Similar to our CIFAR experiments, we find that for networks with standard convolutions, *i.e.*, ResNet50, the lower the precision value the better accuracy is. Thus, the selected precision is

Table 3: ImageNet results for *DualPrecision*. All the activations are quantized to 8 bit. We note that the results for 8 bit are iso-accurate with their full-precision counterparts, which makes 8 bit a strong baseline.

| Networks | Methods | Top-1 (%) | $\mathcal{C}_{size}$ $(10^6)$ | Top-1 (%) | $\mathcal{C}_{size}$ $(10^6)$ | Top-1 (%) | $\mathcal{C}_{size}$ $(10^6)$ |
|---|---|---|---|---|---|---|---|
| | 8 bit | 71.11 | 63.85 | 74.86 | 102.87 | 76.70 | 204.18 |
| ResNet50 | Flexible (Wang et al. (2019)) | 74.30 | 63.60 | 76.04 | 102.90 | 77.23 | 204.18 |
| | DualPrecision (Ours) | **75.44** | 63.13 | **76.70** | 102.83 | **77.58** | 204.08 |
| | 8 bit | 52.17 | 11.99 | 64.39 | 15.44 | 71.73 | 27.50 |
| MobileNetV2 | Flexible (Wang et al. (2019)) | 55.20 | 12.10 | 65.00 | 15.54 | 72.13 | 27.71 |
| | DualPrecision (Ours) | **59.51** | 12.15 | **68.01** | 15.53 | **73.91** | 27.56 |

1 bit. On the other hand, for MobileNetV2, we find that 4 bit for standard convolution and 4 bit for depth-wise convolution perform the best. We consider two baselines to benchmark the proposed approach including 8-bit fixed-point and mixed-precision networks (Wang et al. (2019)) with width-multipliers. For mixed-precision, we follow (Wang et al. (2019)) and use a reinforcement learning approach to search for iso-accurate networks (iso- compared to the 8-bit fixed-point models). Then, we use width-multipliers on top of the searched network to obtain models of different sizes. We consider networks of three sizes, *i.e.*, the size of $0.25\times, 0.5\times$ and $1\times$ 8-bit fixed-point models. As shown in Table 3, our proposed simple heuristic outperforms both baselines by a significant margin for both networks considered.

## 7 CONCLUSION

In this work, we discuss the Pareto efficiency of quantized convolutional neural networks (CNNs). We find that a lower weight precision value produces a more accurate network than higher weight precision one when the model size is aligned using a width-multiplier (*i.e.*, growing or shrinking the number of channels proportionally.) for CNNs with standard convolutions. Furthermore, from both theoretical and empirical analyses, we find that the fan-in channel counts per output filter of a convolution layer determine the optimal precision value for that layer, which explains our observed phenomenon that depth-wise convolutions are less quantization-friendly compared to their standard counterparts. Based on our findings, we propose *DualPrecision*, a simple yet effective heuristic for precision selection of a given network. We show empirically that, when applied on ImageNet, *DualPrecision* outperforms the 8-bit fixed-point baseline and prior art in mixed-precision by a significant margin.

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

## A    TRAINING HYPER-PARAMETERS

For CIFAR, we use a learning rate of 0.05, cosine learning rate decay, linear learning rate warmup (from 0 to 0.05) with 5 epochs, batch size of 128, total training epoch of 300, weight decay of $5e^{-4}$, SGD optimizer with Nesterov acceleration and 0.9 momentum.

For ImageNet, we have identical hyper-parameters as CIFAR except for the following hyper-parameters. Batch size of 256, 120 total epochs for MobileNetV2 and 90 for ResNets, weight decay $4e^{-5}$, and 0.1 label smoothing.

## B    CLIPPING POINT FOR QUANTIZATION-AWARE TRAINING

As mentioned earlier, $\boldsymbol{a} \in \mathbb{R}^{C_{out}}$ denotes the vector of clipping factors which is selected to minimize $\|Q(\boldsymbol{W}_{i,:}) - \boldsymbol{W}_{i,:}\|_2^2$ by assuming $\boldsymbol{W}_{i,:} \sim \mathcal{N}(0, \sigma^2 \boldsymbol{I})$. More specifically, we run simulations for weights drawn from a zero-mean Gaussian distribution with several variances and identify the best $a_i^* = \arg\min_{a_i} \|Q_{a_i}(\boldsymbol{W}_{i,:}) - \boldsymbol{W}_{i,:}\|_2^2$ empirically. According to our simulation, we find that one can infer $a_i$ from the sample mean $|\bar{\boldsymbol{W}}_{i,:}|$, which is shown in Figure 5. As a result, for the different precision values considered, we find $c = \frac{|\bar{\boldsymbol{W}}_{i,:}|}{a_i^*}$ via simulation and use the obtained $c$ to calculate $a_i$ on-the-fly throughout training.

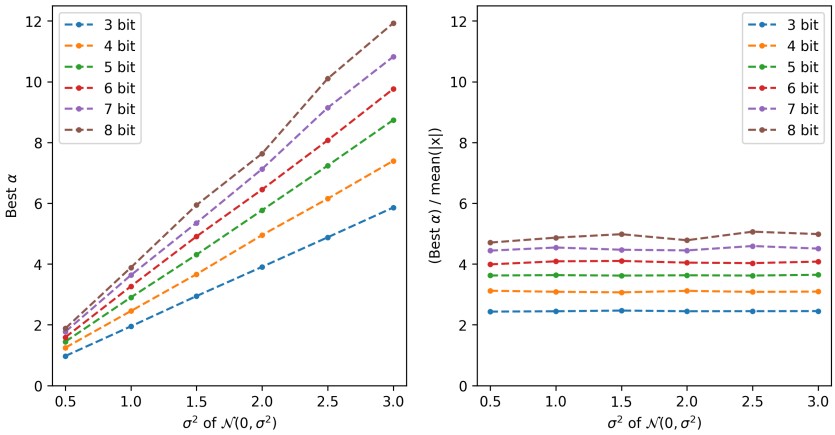

Figure 5: Finding best $a_i$ for different precision values empirically through simulation using Gaussian with various $\sigma^2$.

## C    NETWORK ARCHITECTURES

For the experiments in Section 4, the ResNets used are detailed in Table 4. Specifically, for the points in Figure 1a, we consider ResNet20 to ResNet56 with width-multipliers of $0.5\times, 1\times, 1.5\times$, and $2\times$ for the 4-bit case. Based on these values, we consider additional width-multipliers $2.4\times$ and $2.8\times$ for the 2-bit case and $2.5\times, 3\times, 3.5\times$, and $3.9\times$ for the 1-bit case. We note that the right-most points in Figure 1a is a $10\times$ ResNet26 for the 4 bit case. On the other hand, VGG11 is detailed in Table 6 for which we consider width-multipliers from $0.25\times$ to $2\times$ with a step of 0.25 for the 4 bit case (blue dots in Figure 1b). The architecture of MobileNetV2 used in the CIFAR-100 experiments follows the original MobileNetV2 (Table 2 in Sandler et al. (2018)) but we change the stride of all the bottleneck blocks to 1 except for the fifth bottleneck block, which has a stride of 2. As a result, we down-sample the image twice in total, which resembles the ResNet design for the CIFAR experiments (He et al. (2016)). Similar to VGG11, we consider width-multipliers from $0.25\times$ to $2\times$ with a step of 0.25 for MobileNetV2 for the 4 bit case (blue dots in Figure 1c).

Table 4: ResNet20 to ResNet56

| LAYERS | 20 | 26 | 32 | 38 | 44 | 50 | 56 |
|---|---|---|---|---|---|---|---|
| STEM | CONV2D (16,3,3) STRIDE 1 | | | | | | |
| STAGE 1 | $3 \times \begin{cases} \text{CONV2D}(16,3,3) \text{ STRIDE } 1 \\ \text{CONV2D}(16,3,3) \text{ STRIDE } 1 \end{cases}$ | $4\times$ | $5\times$ | $6\times$ | $7\times$ | $8\times$ | $9\times$ |
| STAGE 2 | $3 \times \begin{cases} \text{CONV2D}(32,3,3) \text{ STRIDE } 2 \\ \text{CONV2D}(32,3,3) \text{ STRIDE } 1 \end{cases}$ | $4\times$ | $5\times$ | $6\times$ | $7\times$ | $8\times$ | $9\times$ |
| STAGE 3 | $3 \times \begin{cases} \text{CONV2D}(64,3,3) \text{ STRIDE } 2 \\ \text{CONV2D}(64,3,3) \text{ STRIDE } 1 \end{cases}$ | $4\times$ | $5\times$ | $6\times$ | $7\times$ | $8\times$ | $9\times$ |

Table 5: Inv-ResNet26

| STEM | CONV2D (16,3,3) STRIDE 1 |
|---|---|
| STAGE 1 | $4 \times \begin{cases} \text{CONV2D}(16 \times 6, 1, 1) \text{ STRIDE } 1 \\ \text{DWCONV2D}(16 \times 6, 3, 3) \text{ STRIDE } 1 \\ \text{CONV2D}(16, 1, 1) \text{ STRIDE } 1 \end{cases}$ |
| STAGE 2 | $4 \times \begin{cases} \text{CONV2D}(32 \times 6, 1, 1) \text{ STRIDE } 1 \\ \text{DWCONV2D}(32 \times 6, 3, 3) \text{ STRIDE } 2 \\ \text{CONV2D}(32, 1, 1) \text{ STRIDE } 1 \end{cases}$ |
| STAGE 3 | $4 \times \begin{cases} \text{CONV2D}(64 \times 6, 1, 1) \text{ STRIDE } 1 \\ \text{DWCONV2D}(64 \times 6, 3, 3) \text{ STRIDE } 2 \\ \text{CONV2D}(64, 1, 1) \text{ STRIDE } 1 \end{cases}$ |

## D  PROOF FOR PROPOSITION 5.1

Based on the definition of variance, we have:

$$\text{Var}(\frac{1}{d} \sum_{i=1}^{d} |\boldsymbol{w}_i|) := \mathbb{E}\left[ \left( \frac{1}{d} \sum_{i=1}^{d} |\boldsymbol{w}_i| \right)^2 - \left( \mathbb{E}\frac{1}{d} \sum_{i=1}^{d} |\boldsymbol{w}_i| \right)^2 \right]$$

$$= \mathbb{E}\left[ \left( \frac{1}{d} \sum_{i=1}^{d} |\boldsymbol{w}_i| \right)^2 - \frac{2\sigma^2}{\pi} \right]$$

$$= \frac{1}{d^2} \mathbb{E}\left( \sum_{i=1}^{d} |\boldsymbol{w}_i| \right)^2 - \frac{2\sigma^2}{\pi}$$

$$= \frac{\sigma^2}{d} + \frac{d-1}{d} \rho \sigma^2 - \frac{2\sigma^2}{\pi}.$$

## E  GRID SEARCH ON IMAGENET

From Table 7, we can observe a trend similar to the CIFAR-100 experiments, *i.e.*, for networks without depth-wise convolutions, the lower precision the better, and for networks with depth-wise convolutions, there are sweet spots for depth-wise convolution and other convolutions. Specifically, the final precision value selected for MobileNetV2 is 4 bit for both depth-wise convolutions and standard convolutions. On the other hand, the selected precision value for ResNet50 is 1 bit.

Table 6: VGGs

| VGG11 | VARIANT A | VARIANT B | VARIANT C |
|---|---|---|---|
| CONV2D (64,3,3) | | | |
| MAXPOOLING | | | |
| CONV2D (128,3,3) | $\begin{cases} \text{CONV2D}(128,1,1) \\ \text{DWCONV2D}(128,3,3) \end{cases}$ | $\begin{cases} \text{CONV2D}(128,1,1) \\ \text{DWCONV2D}(128,3,3) \end{cases}$ | $\begin{cases} \text{CONV2D}(128,1,1) \\ \text{DWCONV2D}(128,3,3) \end{cases}$ |
| MAXPOOLING | | | |
| CONV2D (256,3,3) | CONV2D (256,3,3) | $\begin{cases} \text{CONV2D}(256,1,1) \\ \text{DWCONV2D}(256,3,3) \end{cases}$ | $\begin{cases} \text{CONV2D}(256,1,1) \\ \text{DWCONV2D}(256,3,3) \end{cases}$ |
| CONV2D (256,3,3) | CONV2D (256,3,3) | $\begin{cases} \text{CONV2D}(256,1,1) \\ \text{DWCONV2D}(256,3,3) \end{cases}$ | $\begin{cases} \text{CONV2D}(256,1,1) \\ \text{DWCONV2D}(256,3,3) \end{cases}$ |
| MAXPOOLING | | | |
| CONV2D (512,3,3) | CONV2D (512,3,3) | $\begin{cases} \text{CONV2D}(512,1,1) \\ \text{DWCONV2D}(512,3,3) \end{cases}$ | $\begin{cases} \text{CONV2D}(512,1,1) \\ \text{DWCONV2D}(512,3,3) \end{cases}$ |
| CONV2D (512,3,3) | CONV2D (512,3,3) | CONV2D (512,3,3) | $\begin{cases} \text{CONV2D}(512,1,1) \\ \text{DWCONV2D}(512,3,3) \end{cases}$ |
| MAXPOOLING | | | |
| CONV2D (512,3,3) | CONV2D (512,3,3) | CONV2D (512,3,3) | $\begin{cases} \text{CONV2D}(512,1,1) \\ \text{DWCONV2D}(512,3,3) \end{cases}$ |
| CONV2D (512,3,3) | CONV2D (512,3,3) | CONV2D (512,3,3) | $\begin{cases} \text{CONV2D}(512,1,1) \\ \text{DWCONV2D}(512,3,3) \end{cases}$ |
| MAXPOOLING | | | |

Table 7: Grid search of DualPrecision for MobileNetV2 and ResNet50 with *the model size aligned to the* $0.25\times$ *8 bit models*. Each cell reports the top-1 accuracy of the corresponding model on ImageNet.

| PRECISION FOR CONVS \ DWCONVS | MOBILENETV2 | | | | RESNET50 |
|---|---|---|---|---|---|
| | 8 BIT | 4 BIT | 2 BIT | 1 BIT | NONE |
| 8 BIT | 52.17 | 53.89 | 50.51 | 48.78 | 71.11 |
| 4 BIT | 56.84 | **59.51** | 57.37 | 55.91 | 74.65 |
| 2 BIT | 53.89 | 57.10 | 55.26 | 54.04 | 75.12 |
| 1 BIT | 54.82 | 58.16 | 56.90 | 55.82 | **75.44** |

# F    MEMORY FOOTPRINT FOR INFERENCE

We calculate and report the memory footprint needed for the proposed DualPrecision models and the baseline 8-bit models to do inference with a single image per batch. Specifically, the memory footprint of inference equals the largest input feature maps plus the largest output feature maps plus the weight sizes for the entire network. As shown in Figure 6, DualPrecision outperforms the baseline. That is, considering the streaming inference setting (a single image per batch), DualPrecision requires less memory to achieve equal accurate results compared to the 8-bit models.

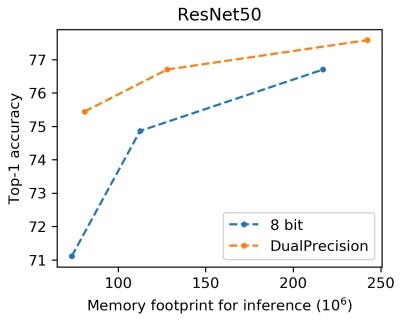

(a) Memory footprint for ResNet50.

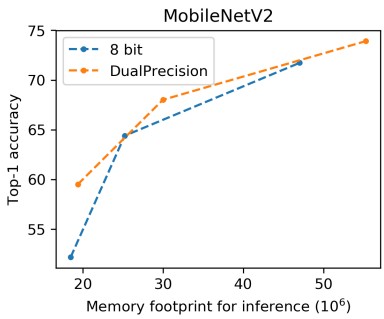

(b) Memory footprint for MobileNetV2.

Figure 6: Memory footprint needed for inference under the single-image-per-batch setting.

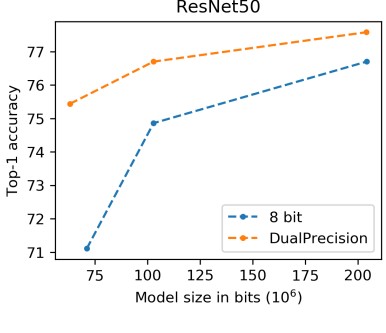

(a) Model size for ResNet50.

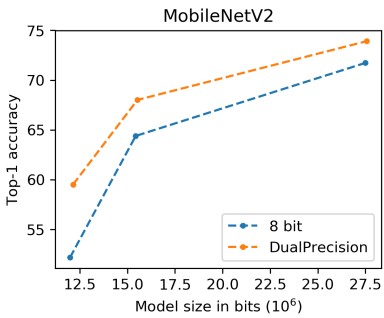

(b) Model size for MobileNetV2.

Figure 7: The visualization of Table 3.

