# OpenReview forum: "On the Pareto Efficiency of Quantized CNN"
_ICLR.cc/2020/Conference — Reject_

### Official Review · AnonReviewer2 · 2019-10-23
**Official Blind Review #2**

**Rating:** 3

**Review:**


This paper studies the accuracy vs model-size trade-off of quantized CNNs under different channel width multipliers. The authors demonstrated that while all-to-all convolution works well under low bit settings, depthwise conv needs a different sweet spot. The authors then proceed to use the insight to design quantized cnns that have two different schemes for depthwise and normal conv.


Strength

The paper is well written and motivated. By adding network width to the search space,  using the simple heuristics, the authors provide a better results than previously DRL based search method.

Weakness

One of my main concerns is the direct usage of total number of bits as an equivalent measurement between models. While it is useful to measure the storage cost for weights. The choices of bit width will likely affect the computing cost in a non-trivial way, depending on the target hardware platform. It is unclear whether equal model size would mean equal inference latency in practice (most likely they would not be). Providing empirical implementations of these models will shed light into this question.

These weakness makes it a borderline paper.

Question:

How do you handle batchnorm layer, do you use floating points?

How many bits did you use for accumulating the results?

Update after rebuttal:

I have read the authors' response. I would like to keep my current review as it is. Also note that the authors uses floating pt for batchnorm and 32 bit for accumulation, such additional cost might out-weights the benefit of choosing ultra low bits in the low bits regime, making the study less relevant for practice reasons




**Experience Assessment:**

I have read many papers in this area.

**Review Assessment: Checking Correctness Of Derivations And Theory:**

I assessed the sensibility of the derivations and theory.

**Review Assessment: Checking Correctness Of Experiments:**

I carefully checked the experiments.

**Review Assessment: Thoroughness In Paper Reading:**

I read the paper thoroughly.

---

> ### Author Response · Authors · 2019-11-09
> **Response to Reviewer #2**
>
> We thank the reviewer for their feedback and for finding our paper well-written and motivated.
>
> Regarding your main concern, we agree that it is non-trivial to see if equal model size reflects compute (in terms of latency and/or energy) since it depends on the application scenario and software/hardware implementation. However, our insights and good results from the model size standpoint set a strong motivation for future study to target a specific application scenario and research on hardware acceleration. We argue that this does not constitute a weakness for our paper since there are scenarios where model sizes is important. From the perspective of information theory and Minimum Description Length (MDL) principle [1,2], our results show that by trading weight precision values with the number of channels, one can achieve a smaller description length for the model with equal accuracy, which is more preferable based on the MDL principle. Moreover, considering edge devices that are deployed for streaming inference scenarios (single image per batch), our analyses in the updated manuscript (Appendix F and Figure 6) show lower memory footprint is needed for the proposed model to achieve equally accurate results compared to the baseline. We argue that for streaming inference applications that run on IoT devices, model size is an important factor due to their limited RAM.
>
> Beyond the above argument for the efficacy of DualPrecision, we would like to re-iterate that our contributions are more than the proposed DualPrecision method. Specifically, the following three contributions are non-trivial, novel, and can be built upon for future study:
> -	We are the first to show that lower precision weight values outperform higher precision weight values in a Pareto sense (accuracy vs. model size) for networks with standard convolutions. This is intriguing since it implies that scaling up (in terms of model size) along the channel count dimension is more effective for accuracy than the precision value dimension. This finding can lead to follow-up works trying to understand or exploit this phenomenon.
>
> -	We are the first to show that the fan-in channel counts per output filter for a convolution layer determine the effectiveness of accuracy improvement along the weight precision dimension and provide both theoretical and empirical reasoning for this. This finding is useful for future works that are interested in optimizing the neural architecture regarding both channel counts and weight precision as we show what might affect the effectiveness of weight precision scaling.
>
> -	We are the first to show that with a simple scaling rule, one can achieve a more accurate model (given the same model size) even compared to mixed-precision prior art that uses DRL to search for layer-wise weight precision values. Moreover, the results are validated on the large-scale dataset, i.e., ImageNet. This is a manifestation of our two previous findings.
>
> We hope the reviewer can take into account the above-listed contributions and their potential impacts when making the final recommendation.
>
> Regarding the questions, we use floating-point for batch norm. The accumulation of results is done in 32 bits (We simulate the quantization process in PyTorch according to Figure C.4 in [3]).
>
> [1] Blier, Léonard, and Yann Ollivier. "The description length of deep learning models." NeurIPS 2018.
> [2] Havasi, Marton, Robert Peharz, and José Miguel Hernández-Lobato. "Minimal Random Code Learning: Getting Bits Back from Compressed Model Parameters." ICLR 2019.
> [3] Benoit Jacob, Skirmantas  Kligys,  Bo  Chen,  Menglong  Zhu,  Matthew  Tang,  Andrew  Howard,Hartwig Adam, and Dmitry Kalenichenko. Quantization and training of neural networks for effi-cient integer-arithmetic-only inference. InThe IEEE Conference on Computer Vision and PatternRecognition (CVPR), June 2018.

---

### Official Review · AnonReviewer3 · 2019-10-24
**Official Blind Review #3**

**Rating:** 6

**Review:**

The author studies the quantization strategy of CNNs in terms of Pareto Efficiency. Through a series of experiments with three standard CNN models (ResNet, VGG11, MobileNetV2), the authors demonstrated that lower precision value can be better than high precision values in term of Pareto efficiency under the iso-model size scenario. They also study cases with and without depth-wise convolution, and propose a new quantization method, DualPrecision. DualPrecision empirically outperformed 8-bit quantization and flexible quantization methods on ImageNet.

Comment:
I am not at all familiar with quantization methods, therefore no knowledge of relevant related works. If, however, the authors did a thorough job of surveying related works and chose sensible baselines, I think the experiments demonstrate the usefulness of the new DualPrecision technique.

**Experience Assessment:**

I do not know much about this area.

**Review Assessment: Checking Correctness Of Derivations And Theory:**

I did not assess the derivations or theory.

**Review Assessment: Checking Correctness Of Experiments:**

I assessed the sensibility of the experiments.

**Review Assessment: Thoroughness In Paper Reading:**

I made a quick assessment of this paper.

---

> ### Author Response · Authors · 2019-11-09
> **Response to Reviewer #3**
>
> We thank the reviewer for your feedback and for appreciating our work.
>
> We have updated our manuscript so that it states our contributions explicitly. While our proposed DualPrecision is simple yet effective, we would like to emphasize that it is only one of our contributions. For the reviewer’s convenience, we re-iterate our main contributions below:
> Overall, we systematically analyze the model size and accuracy trade-offs considering both weight precision values and the number of channels for various modern networks architectures (variants of ResNet, VGG, and MobileNet) and datasets (CIFAR and ImageNet) and have the following non-trivial and novel contributions:
> -	We are the first to show that lower precision weight values outperform higher precision weight values in a Pareto sense (accuracy vs. model size) for networks with standard convolutions. This is intriguing since it implies that scaling up (in terms of model size) along the channel count dimension is more effective for accuracy than the precision value dimension. This finding can lead to follow-up works trying to understand or exploit this phenomenon.
>
> -	We are the first to show that the fan-in channel counts per output filter for a convolution layer determine the effectiveness of accuracy improvement along the weight precision dimension and provide both theoretical and empirical reasoning for this. This finding is useful for future works that are interested in optimizing the neural architecture regarding both channel counts and weight precision as we show what might affect the effectiveness of weight precision scaling.
>
> -	We are the first to show that with a simple scaling rule, one can achieve a more accurate model (given the same model size) even compared to mixed-precision prior art that uses DRL to search for layer-wise weight precision values. Moreover, the results are validated on the large-scale dataset, i.e., ImageNet. This is a manifestation of our two previous findings.
>
> We hope the reviewer can take into account the above-listed contributions and their potential impacts when making the final recommendation.

---

### Official Review · AnonReviewer1 · 2019-10-27
**Official Blind Review #1**

**Rating:** 3

**Review:**

This paper investigated the Pareto efficiency of quantized convolutional neural networks (CNNs).
The authors pointed out that  when the number of channels of a network are changed in an iso-model size scenario, the lower precision value can Pareto dominate the higher ones for CNNs with standard convolutions. Furthermore, the authors showed that depth-wise convolutions are quantization-unfriendly when going low- precision even in quantization-aware training. We further provide theoretical insights for it and show that the number of input channels per output filter affects the Pareto optimal precision value. The authors then proposed DualPrecision, a simple yet effective heuristic for precision selection of a given network. When applied on ImageNet, DualPrecision outperforms the 8 bit fixed point baseline and prior art in mixed-precision by a significant margin.

1. The wording issues
- The paper used words like "iso-model" and "Pareto" at many parts of the draft, making it quite hard to read..

2. The experiments seem fair. However, the improvement seems quite marginal
The proposed method improved the baseline by 0.9% and 2.2% in top-1 accuracy on ImageNet for ResNet50 and MobileNetV2, respectively.

I have to admit that I am unfamiliar with this particular topic and only knows a few classic papers, like for example Xnor-net: Imagenet classification using binary convolutional neural networks

**Experience Assessment:**

I do not know much about this area.

**Review Assessment: Checking Correctness Of Derivations And Theory:**

I assessed the sensibility of the derivations and theory.

**Review Assessment: Checking Correctness Of Experiments:**

I assessed the sensibility of the experiments.

**Review Assessment: Thoroughness In Paper Reading:**

I read the paper at least twice and used my best judgement in assessing the paper.

---

> ### Author Response · Authors · 2019-11-09
> **Response to Reviewer #1**
>
> We thank the reviewer for the feedback.
>
> Regarding the wording choices, we’ve embraced your suggestions and adjusted our manuscript accordingly (marked in red). Specifically, we added a definition for Pareto domination for better understanding and remove the usage of iso-model size when possible.
>
> We appreciate the reviewer’s concern, to argue for the significance of our results, we’ve gathered the recently accepted papers in major conferences (ICLR’19 and NeurIPS’19) and report their top-1 accuracy improvements on the ImageNet dataset.
>
> [1] improves upon its prior art by 0.6% (74 - > 74.6)
> [2] improves upon the baseline by 0.3% (75.3 -> 75.6)
> [3] improves upon its prior art by 0.8% (72.5 -> 73.3)
> [4] improves upon the baseline by 0.97% (76.34 -> 77.31)
> [5] improves upon its prior art by 0.23% (74.95 -> 75.18)
> [6] improves upon its prior art by 1.49% (58.35 -> 59.84)
> [7] improves upon its prior art by 0.7% (75.5 -> 76.2)
> [8] improves upon its prior art by 1.5% (74.5 -> 76.0)
>
> Hence, we argue our improvements, i.e., 0.9% and 2.2%, are significant. To see our improvements from the model size perspective, our ResNet50 results in Table 3 achieve 2x and 8x model size reduction without accuracy degradation compared to the 8-bit and floating-point model, respectively. Moreover, we provide larger improvements in the lower model size regimes (Table 3).
>
> Lastly, we thank the reviewer and appreciate the reviewer’s time even when this paper does not match your particular area of expertise. We improved the manuscript and explicitly point out the contributions and the novelty of this paper, which are summarized below:
>
> Overall, we systematically analyze the model size and accuracy trade-offs considering both weight precision values and the number of channels for various modern networks architectures (variants of ResNet, VGG, and MobileNet) and datasets (CIFAR and ImageNet) and have the following non-trivial and novel contributions:
> -	We are the first to show that lower precision weight values outperform higher precision weight values in a Pareto sense (accuracy vs. model size) for networks with standard convolutions. This is intriguing since it implies that scaling up (in terms of model size) along the channel count dimension is more effective for accuracy than the precision value dimension. This finding can lead to follow-up works trying to understand or exploit this phenomenon.
>
> -	We are the first to show that the fan-in channel counts per output filter for a convolution layer determine the effectiveness of accuracy improvement along the weight precision dimension and provide both theoretical and empirical reasoning for this. This finding is useful for future works that are interested in optimizing the neural architecture regarding both channel counts and weight precision as we show what might affect the effectiveness of weight precision scaling.
>
> -	We are the first to show that with a simple scaling rule, one can achieve a more accurate model (given the same model size) even compared to mixed-precision prior art that uses DRL to search for layer-wise weight precision values. Moreover, the results are validated on the large-scale dataset, i.e., ImageNet. This is a manifestation of our two previous findings.
>
> We hope the reviewer can take into account the above-listed contributions and their potential impacts when making the final recommendation.
>
> [1] Cai, Han, Ligeng Zhu, and Song Han. "Proxylessnas: Direct neural architecture search on target task and hardware." ICLR 2019.
> [2] Chen, Hao-Yun, et al. "Complement Objective Training." ICLR 2019.
> [3] Liu, Hanxiao, Karen Simonyan, and Yiming Yang. "Darts: Differentiable architecture search." ICLR 2019.
> [4] Chen, Chun-Fu, et al. "Big-little net: An efficient multi-scale feature representation for visual and speech recognition." ICLR 2019.
> [5] You, Zhonghui, et al. "Gate decorator: Global filter pruning method for accelerating deep convolutional neural networks." NeurIPS 2019.
> [6] Chen, Shangyu, et al. “MetaQuant: Learning to Quantize by Learning to Penetrate Non-differentiable Quantization.” NeurIPS 2019.
> [7] Dong, Xuanyi, and Yi Yang. "Network Pruning via Transformable Architecture Search." NeurIPS 2019.
> [8] Nayman, Niv, et al. "XNAS: Neural Architecture Search with Expert Advice." NeurIPS 2019.

---

### Decision · Program_Chairs · 2019-12-19

**Decision:**

Reject

**Comment:**

This paper studies the trade-off between the model size and quantization levels in quantized CNNs by varying different channel width multipliers. The paper is well  motivated and draws interesting observations but can be improved in terms of evaluation. It is a borderline case and rejection is made due to the high competition.